# Placental Types and Effective Perinatal Management of Vasa Previa: Lessons from 55 Cases in a Single Institution

**DOI:** 10.3390/diagnostics11081369

**Published:** 2021-07-29

**Authors:** Daisuke Tachibana, Takuya Misugi, Ritsuko K. Pooh, Kohei Kitada, Yasushi Kurihara, Mie Tahara, Akihiro Hamuro, Akemi Nakano, Masayasu Koyama

**Affiliations:** 1Department of Obstetrics and Gynecology, Graduate School of Medicine, Osaka City University, Osaka 545-8585, Japan; misutaku1975@infoseek.jp (T.M.); kafukafu0404@yahoo.co.jp (K.K.); kurikuri_1011@yahoo.co.jp (Y.K.); rxv13436@nifty.ne.jp (M.T.); hamuroa@med.osaka-cu.ac.jp (A.H.); m2037746@med.osaka-cu.ac.jp (A.N.); masayasukoyama@gmail.com (M.K.); 2Fetal Diagnostic Center, Fetal Brain Center, CRIFM Clinical Research Institute of Fetal Medicine PMC, Osaka 543-0001, Japan; ritsuko.pooh.brain@fetal-medicine-pooh.jp

**Keywords:** vasa previa, diagnosis, management, cervical length, tocolysis, cervical cerclage, the Ward technique

## Abstract

Background: We aimed to identify clinical characteristics and outcomes for each placental type of vasa previa (VP). Methods: Placental types of vasa previa were defined as follows: Type 1, vasa previa with velamentous cord insertion and non-type 1, vasa previa with a multilobed or succenturiate placenta and vasa previa with vessels branching out from the placental surface and returning to the placental cotyledons. Results: A total of 55 cases of vasa previa were included in this study, with 35 cases of type 1 and 20 cases of non-type 1. Vasa previa with type 1 showed a significantly higher association with assisted reproductive technology, compared with non-type 1 (*p* = 0.024, 60.0% and 25.0%, respectively). The diagnosis was significantly earlier in the type 1 group than in the non-Type 1 group (*p* = 0.027, 21.4 weeks and 28.6 weeks, respectively). Moreover, the Ward technique for anterior placentation to avoid injury of the placenta and/or fetal vessels was more frequently required in non-type 1 cases (*p* < 0.001, 60.0%, compared with 14.3% for type 1). Conclusion: The concept of defining placental types of vasa previa will provide useful information for the screening of this serious complication, improve its clinical management and operative strategy, and achieve more preferable perinatal outcomes.

## 1. Introduction

Vasa previa (VP) is a serious obstetric complication that can result in fetal exsanguination due to the laceration of unprotected fetal vessels which run through the membrane in close proximity to the internal cervical os [1]. The incidence of VP has been estimated to be one in 2500 pregnancies [2], and morbidity and mortality are very high if the diagnosis of the condition is uncertain before the rupture of membranes and labor onset [2].

Recently, obstetricians seem to be more cautious in the screening of umbilical cord insertion, owing to the increased attention and accumulated knowledge of VP, including risk factors such as velamentous cord insertion, the presence of second-trimester placenta previa, and pregnancy by assisted reproductive technology (ART) [3,4,5,6]. In fact, the recent awareness of VP detection and management has greatly improved neonatal mortality and morbidity [7,8,9]. However, in the obstetrical practice, the diagnosis of VP remains a challenge, especially in cases where fetal vessels branch out from the placental surface and return to the placental cotyledons in a “boomerang orbit” [6,10] or in cases where umbilical veins have a slow blood velocity adjacent to maternal blood perfusing through the placental intervillous space [11]. Additionally, the management of VP cases with premature uterine contractions or progressive cervical shortening far from term, as well as peri-operative strategies such as a safe approach to the fetus without injuring the placenta and/or velamentous vessels, have not yet been fully discussed.

Our aims in this study were (1) to identify clinical characteristic and outcomes according to each placental type of VP and (2) to provide a more in-depth depiction of the essential clinical requirements and operative techniques for the effective management of VP cases and to describe their own particular obstetrical challenges.

## 2. Materials and Methods

### 2.1. Study Design and Patients

The medical records of patients complicated with VP and who had undergone perinatal management at Osaka City University Hospital between March of 2010 and February of 2021 were retrospectively reviewed. All patients gave their informed written consent, and the study protocol was approved by the Institutional Review Board (No. 2020-051).

Osaka City University Hospital is a tertiary medical center. Some pregnant women visit our institution out of to their wish for obstetrical management, and others are referred by local obstetricians due to the suspicion of high-risk pregnancy at any gestational week (GW). The routine ultrasound scanning of cervical length and/or umbilical cord insertion for patients visiting from the first trimester is as follows: after confirmation of GW by measuring the crown-rump length, a transvaginal ultrasound (TV-US) is performed every 4 weeks until the 24th GW and, thereafter, TV-US is used for those cases suspected of low placentation. In cases with no abnormal TV-US finding, subsequent use of the device is left to each obstetrician’s discretion. A trans-abdominal ultrasound (TA-US) for the detection of umbilical cord insertion is performed between the 16th and 24th GW. In those cases with suspected VP, TV-US is taken not only in the sagittal plane but also in the non-sagittal plane. VP is defined as fetal vessels running within 2 cm of the internal cervical ostium. If the fetal vessels are more than 2 cm apart from the internal cervical ostium after the diagnosis of VP, those cases are defined as resolution. Low-lying placenta is defined when the placental edge is found to be within 2 cm from the internal cervical ostium. For those same cases where it is difficult to distinguish whether veins have originated from the fetal umbilical vein or from the maternal blood perfusing the placental vascular bed, the Valsalva maneuver is applied. If the veins are of maternal origin, the blood flow will show fluctuation simultaneously with maternal breath [11]. Three cases reported previously were included in this study [11].

The patients in this study, who were diagnosed with VP before 30th GW, were scheduled to be hospitalized between 30th–31st GW. Those with symptomatic uterine contraction and/or shortened cervix were hospitalized at any GW. After hospitalization, routine cardiotocogram (CTG) monitoring was performed twice a day (for about 40 min) for asymptomatic patients, and tocolytic treatment was started if any uterine contraction was observed within 10 to 15 min, even without pain or cervical shortening. The routine administration of antenatal steroids for fetal lung maturation was refrained from for stable cases. Cervical cerclage was considered for those cases with a shortened cervix less than 25 mm and recognized before 25th GW. A cesarean section (CS) was planned between 35–36th GW for cases where expectant management was possible. If the VP diagnosis was made after 30th GW, the patient was admitted soon after the diagnosis and, if the diagnosis was made after 36th GW, the CS was scheduled immediately. Before the CS, the operation team, including well-experienced obstetricians, performed meticulous pre-operational mapping for placental and fetal vessels with TA- and TV-US, especially at the site where the uterine incision was to be made. For those cases with an anteriorly located placenta covering the incision site, the Ward technique was planned to avoid injury to the placenta (Figure 1) [12,13]. Routinely, in cases where fetal vessels course just beneath the anterior wall, we first hold and lift the fetal presenting part with the intact membrane after the lateral elongation of the myometrial incision. The membrane is then ruptured, thus avoiding injury to the fetal vessels [14]. The final determination of the particular VP type is based on the findings at delivery and subsequent macroscopic examination of the placenta. In this study, the placental types for VP were evaluated as follows: type 1, a velamentous cord insertion into the placenta [1]; type 2, a multilobed or succenturiate placenta with fetal vessels connecting the placental lobes [1]; type 3, vessels branching out from the placental surface and returning to the placental cotyledons in a “boomerang orbit” [6,10]. The two cases with type 3 VP reported previously were also included in this study [10]. The VP patients were divided into two groups for statistical analysis to elucidate the clinical differences between the groups: the type 1 group included type 1 VP patients, and the non-Type 1 group included those patients with either type 2 or type 3 VP placentas. The standard deviation of birth weight was calculated using software from the Japanese Society for Pediatric Endocrinology (http://jspe.umin.jp/taikakubirthlongcrossv1.xlsx; downloaded on 1 March 2021).

### 2.2. Statistical Analysis

Continuous variables were expressed as median (range), and categorical variables were expressed as numbers (%). Differences between the type 1 group and the non-type 1 group were studied using the Chi-squared test or Fisher’s exact test for categorical variables and Mann–Whitney U test for continuous variables. A *p*-value < 0.05 was considered statistically significant. Analysis was carried out with the BellCurve for Excel (Social Survey Research Information Co., Ltd., Tokyo, Japan).

## 3. Results

### 3.1. Characteristics of the Patients

During the study period, there were 55 women, including three monochorionic diamniotic twin pregnancies, which were diagnosed as having VP. Maternal characteristics by placental group are shown in Table 1. There were 35 cases of type 1 and 20 cases of non-type 1 (13 cases of type 2 and 7 cases of type 3 placentas). Three women with monochorionic diamniotic twin pregnancies were included in the type 1 group, and two of these cases were conceived by ART. The number of women who conceived via ART was significantly higher in the type 1 group than in the non-type 1 group. Thirty-two cases were referred to our hospital due to VP.

The characteristics of diagnosis and placental location are shown in Table 2. No cases were recognized as VP after delivery. The GW of diagnosis was significantly earlier in the type 1 group than in the non-type 1 group (*p* = 0.027). Among all of the cases, 10 cases (18.2%) were diagnosed after the 32nd GW. Thirty-seven cases (67.3%) were diagnosed by the non-sagittal view with TV-US, and the Valsalva maneuver was applied in 11 of these cases (20.0%) to obtain a better differentiation between maternal and fetal veins. There were 34 cases (61.8%) with low-lying placenta at the time of diagnosis. At the time of the CS, low-lying placenta was more frequently observed in the non-type 1 group (*p* = 0.026).

### 3.2. Clinical Management and Operative Outcomes

Table 3 shows the results of management and operative outcomes. In 31 cases (56.4%), including three twin pregnancies, tocolytic therapy was necessary (ritodrine hydrochloride and/or magnesium sulfate), and a steroid for fetal lung maturation was given in 9/13 cases (69.2%) delivered before the 34th GW. An abnormal fetal heart rate pattern without uterine contractions was detected in one case in the type 1 group, and this case exhibited a relatively thick artery near the internal ostium (Figure 2). Cervical cerclage via the McDonald technique was performed in two cases in the type 1 group. One of these cases was a singleton pregnancy at 20th GW and a cervical length of 2.1 cm, delivered by CS at 33rd GW. The other was a monochorionic-diamniotic pregnancy case at 20th GW and a cervical length of 1.7 cm, delivered by CS at 31st GW; this patient had a previous history of preterm singleton delivery at 34th GW. Blood flow of fetal vessels was confirmed post-cerclage in both cases (Figure 3). Resolution was observed in 12 cases (21.8%) in total. An emergent CS was performed in 25 cases with increased uterine contractions and in one case with an indication of hypertensive disorder of pregnancy; none of these cases experienced the premature rupture of membranes. Although the Ward technique was more frequently used in the non-type 1 group, the details as to operative outcome showed no significant difference between the two groups. The transection of the placenta to approach the fetus was not used in any of the cases. All cases underwent a cesarean section, even in those cases of resolution of VP.

### 3.3. Neonatal Outcomes

Neonatal outcomes are presented in Table 4. There was no clinical difference in neonatal outcomes, including the standard deviation of the birth weights between the two groups, and no case necessitated a transfusion for the treatment of anemia caused by exsanguination. There were four cases with congenital disorders: two cases had infantile hemangioma, one case had a single umbilical artery yet otherwise normal anatomy, and one case showed a severe metabolic disorder with hyperammonemia which was normalized within one month, even though the infant later showed epileptic syndrome in infancy.

## 4. Discussion

The percentage of conception by ART was significantly higher in the type 1 VP group. The timing of diagnosis in the non-type 1 group was 7 weeks later than that in the type 1 group. Cervical cerclage was safely performed in two cases and did not trigger uterine contractions and the rupture of the membranes, and this potentially enabled for the delay of an urgent CS. As far as we know, this is the first report to describe cervical cerclage treatment for VP cases. The Ward technique for anterior placenta was significantly more frequently necessitated in the non-type 1 VP cases, possibly due to a significantly higher concomitancy with low-lying placenta at the time of the CS.

It has been argued that the routine screening for VP places considerable demands on both patients and medical resources. The rate of false-positives and false-negatives should also be considered. However, our study showed that 25% of non-type 1 VP patients were diagnosed after the 32nd GW, and some of these cases required advanced techniques in order to detect VP [10,11]. Furthermore, even when the velamentous insertion site was recognized to be relatively distant from the internal cervical ostium, some cases showed a very long, aberrant circumventing vessel running on the membranes covering it (Figure 4). Therefore, we agree with the proposal that TV-US with Doppler imaging should be carefully performed at about 32nd GW in patients who have a second trimester low-lying placenta [15].

Elective hospitalization and the timing of admission for VP patients remains a matter of debate [7,15,16,17,18,19,20,21]. It is obvious that inpatient management will increase the chances to monitor uterine activity and abnormal fetal heart rate patterns, and this situation will therefore increase the rate of medical interventions. In our study, however, the timing of the delivery was comparable to other reports [18,22,23,24], and appropriate intervention seems to have been undertaken to avoid fetal/neonatal exsanguination. The issue of elective hospitalization should be further assessed in prospective studies with a larger patient population.

Catanzarite et al. reported on the use of tocolytics in 67% of singleton VP cases and 95% of twin VP cases, with the first treatment at 30.6 ± 3.2rd GW, and 26.0 ± 3.3rd GW for singleton and twin pregnancies, respectively [7]. Their frequency of tocolytic use was comparable with our results. This high frequency of tocolytic use at both centers might be the result of elective hospitalization, which allowed obstetricians to detect painless uterine activity by routine CTG monitoring. The rate of cervical shortening in VP management is also discussed by Maymon et al. [25]. They reported that the odds of an emergency CS increased by 6.50 (95% confidence interval, 1.02–41.20) for each additional millimeter-per-week decrease in cervical length, and they suggested the clinical use of cervical length scanning with the rate of shortening in the management of VP [25]. More recently, several authors have shown that fetoscopic laser ablation of type 2 VP was technically achievable and brought favorable outcomes [26,27,28,29,30]. However, this treatment can only be applied in bilobed- and multilobed-placentas, which perfuse a relatively small segment of placenta (≦10%) [30]. Taking into consideration that emergent cerclage is associated with longer latency periods and better pregnancy outcomes [31,32,33], our approach to perform cervical cerclage in cases showing a shortened cervix might potentially be an optional treatment for all types of VP in the pre-viability and severe prematurity periods.

It has been rarely discussed as to the appropriate incision site of the uterine wall depending on the location of these vessels. Except in cases where placental invasion into the myometrium was suspected, we did not change the uterine incision site regardless of where the placenta was located and/or the fetal vessels coursed in pre-operational mapping. This technique was first introduced by Ward for cases with anterior placenta previa [12] and has recently shown to be safely applied in a study of a larger population [13]. We consider that the Ward technique for anterior placentation, and partially the ‘en caul’ technique for the anterior coursing of fetal vessels, might be also safely and successfully applied for VP cases.

Morphological anomalies of the placenta and cord insertion in ART pregnancies has been observed for over three decades [14,34,35]. Two speculations are proposed for these conditions: firstly, disorientation of a polarized zygote as the cause of velamentous cord insertion [34] and, secondly, superficial implantation of the zygote and subsequent contact to the opposite uterine wall as the cause of bilobed and succenturiate placentas [35]. Although these theories remain to be further elucidated, our results that 60.0% of type 1 and 25.0% of non-type 1 were conceived via ART might further hint at a mechanistic relationship between VP and ART. In addition, Torpin’s superficial implantation theory might explain the reason why non-Type1 cases tend to be diagnosed at a later GW, where the expansion of uterine volume makes it apparent that the placentas are separately located from each other [36]. Recent advances of genome-wide studies of placental diseases may also add helpful information to elucidate the underlying mechanisms [37,38].

There might be some limitations in our study. The first is the retrospective nature of the study design. The second is that our center is tertiary and, as a result, patients could be referred at any GW, and the diagnostic protocol was not uniform. The third is that our management recommended elective hospitalization routinely without any individual assessment. In addition, we lack the proposal to reduce the burden of screening and prenatal visit, as well as the consequences of abnormal placentation leading to post-operative complications [39].

## 5. Conclusions

Our study suggests that conception by ART has a higher association with type 1 VP and that the diagnosis of non-type 1 VP tends to be made at a significantly later gestational week and as much as 7 weeks. However, excellent outcomes for both mothers and neonates could be achieved by more careful monitoring and adequate measures. Pre-operatively, the ideal operative technique to deliver the fetus while avoiding placental and/or fetal vessel injury should be discussed by all obstetrical team members who are on duty in emergent situations.

## Figures and Tables

**Figure 1 diagnostics-11-01369-f001:**
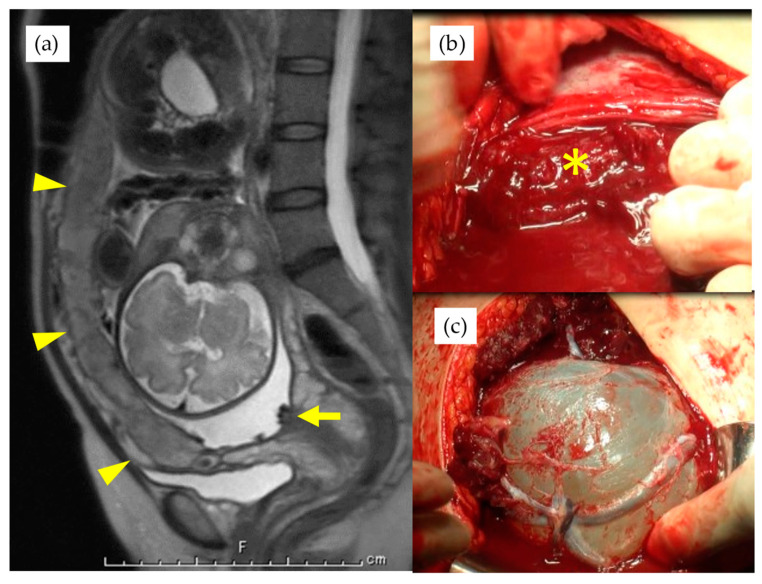
(**a**) magnetic resonance imaging of a case with the placenta covering whole anterior uterine wall, indicated by arrowheads, and arrow indicates fetal vessels covering cervical internal ostium. * indicates placenta after lateral extension of uterine incision (**b**) and the bleeding from the maternal vascular bed. Velamentous fetal vessels are visible on the amniotic membrane after the Ward technique (**c**).

**Figure 2 diagnostics-11-01369-f002:**
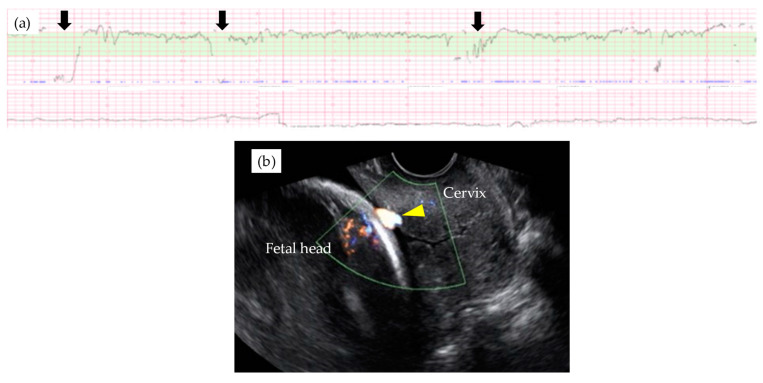
Abnormal fetal heart rate patterns (**a**) in a type 1 VP case with relatively large arterial vasa previa at 29th GW (**b**). Note that fetal heart rate decelerations were observed without uterine contraction (black arrown in Figure 2a) and relatively thick velamentous vessel was identified by the trans-vaginal ultrasound scan (yellow arrow head in Figure 2b).

**Figure 3 diagnostics-11-01369-f003:**
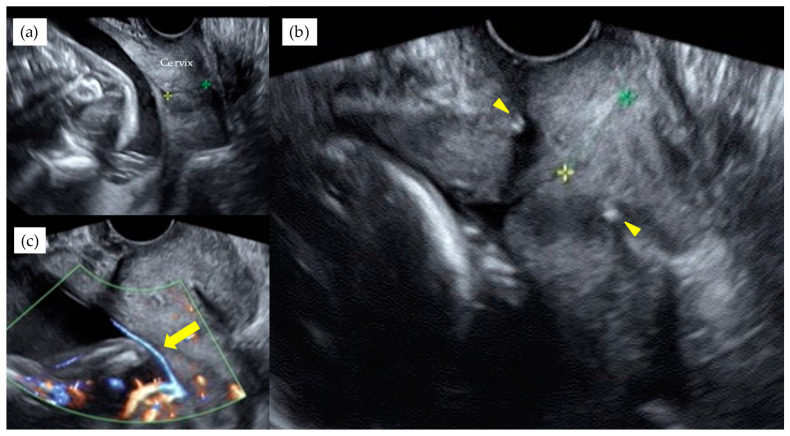
A monochorionic diamniotic pregnancy case of type 1 VP, with previous history of preterm singleton delivery at 34th GW, showed cervical length of 1.7 cm at 20th GW (**a**). Blood flow of fetal vessels were confirmed after cervical cerclage performed. Arrowheads indicate string of cervical cerclage (**b**). Arrows indicate vasa previa with color Doppler image after cervical cerclage (**c**). This patient delivered by CS at 31st GW. The yellow and green stars indicate the cervical length.

**Figure 4 diagnostics-11-01369-f004:**
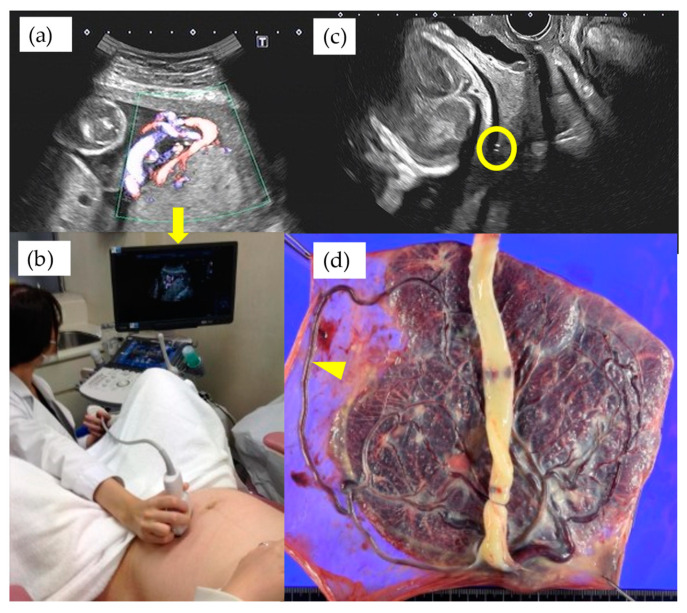
A case of VP at 27th GW showed the velamentous insertion site relatively distant from internal cervical ostium shown with TA-US (**a**) and the hand of examiner positioned at the navel height of the patient to visualize the cord insertion (**b**). Yellow circle indicates the fetal vessel running on the internal cervical ostium detected by TV-US on the same day of examination (**c**) and arrowhead indicates the macroscopic finding of very long aberrant circumventing vessel (**d**). The arrow in Figure 4a indicates the magnified monitor of the ultrasound image of the Figure 4b.

**Table 1 diagnostics-11-01369-t001:** Maternal characteristics of the study.

	Total *n* = 55	Type 1 *n* = 35	Non-Type 1 *n* = 20	*p*-Value
Age	36 (21–49)	38 (21–43)	34 (29–49)	0.759
Gravida	1 (1–6)	1 (1–6)	2 (1–4)	0.308
Parity	0 (0–2)	0 (0–2)	0 (0–1)	0.192
ART, *n* (%)	26 (47.3%)	21 (60.0%)	5 (25.0%)	0.024
Twin pregnancy, *n* (%)	3 (5.5%)	3 (8.6%)	0 (0%)	0.293
Referred after VP diagnosis, *n* (%)	32 (58.2%)	25 (71.4%)	7 (35.0%)	0.019

ART, assisted reproductive technology. Data are given as median (range) or *n* (%).

**Table 2 diagnostics-11-01369-t002:** Characteristics of diagnosis and placental location.

	Total *n* = 55	Type 1 *n* = 35	Non-Type 1 *n* = 20	*p*-Value
Diagnosis of GW	25.1 (18.0–39.0)	21.4 (18.0–39.0)	28.6 (18.3–35.0)	0.027
Diagnosis ≦ 20th GW, *n* (%)	*n* = 16 (29.1%)	*n* = 14 (40.0%)	*n* = 2 (10.0%)	0.029
Diagnosis ≦ 24th GW, *n* (%)	*n* = 26 (47.3%)	*n* = 20 (57.1%)	*n* = 6 (30.0%)	0.097
Diagnosis ≦ 28th GW, *n* (%)	*n* = 34 (61.8%)	*n* = 25 (71.4%)	*n* = 9 (45.0%)	0.099
Diagnosis ≦ 32th GW, *n* (%)	*n* = 45 (81.8%)	*n* = 30 (85.7%)	*n* = 15 (75.0%)	0.530
Non-sagittal plane, *n* (%)	*n* = 37 (67.3%)	*n* = 24 (68.6%)	*n* = 13 (65.0%)	1.000
Valsalva maneuver, *n* (%)	*n* = 11 (20.0%)	*n* = 4 (11.4%)	*n* = 7 (35.0%)	0.076
Low-lying placenta at diagnosis, *n* (%)	*n* = 34 (61.8%)	*n* = 18 (51.4%)	*n* = 16 (80.0%)	0.070
Low-lying placenta at CS, *n* (%)	*n* = 29 (52.7%)	*n* = 14 (40.0%)	*n* = 15 (75.0%)	0.026

GW, gestational week. Data are given as median (range) or *n* (%).

**Table 3 diagnostics-11-01369-t003:** Characteristics of the diagnosis and managements.

	Total *n* = 55	Type 1 *n* = 35	Non-Type 1 *n* = 20	*p*-Value
GW of admission	31.4 (24.3–39.3)	30.8 (24.3–39.3)	31.8 (24.3–35.8)	0.457
Usage of tocolytic agent, *n* (%)	31 (56.4%)	20 (57.1%)	11 (55.0%)	1.000
Duration of tocolysis (days)	5 (0–69)	5 (0–68)	4 (0–69)	0.762
First treatment with tocolytics (GW)	30.9 (24.3–34.6)	30.8 (24.3–34.6)	32.1 (24.3–33.1)	0.580
Steroid administration, *n*/*N* (%)	15/55 (27.3%)	10/35 (28.6%)	5/20 (25.0%)	1.000
Steroid administration for the cases delivered before 34th GW, *n*/*N* (%)	9/13 (69.2%)	8/9 (88.9%)	1/4 (25.0%)	0.052
Abnormal CTG, *n* (%)	1 (1.8%)	1 (2.9%)	0 (0%)	1.000
Cervical cerclage, *n* (%)	2 (3.6%)	2 (5.7%)	0 (0%)	0.529
Resolution, *n* (%)	12 (21.8%)	9 (25.7%)	3 (15%)	0.503
GW of delivery (GW)	35.1 (30.3–39.3)	35.1 (31.0–39.3)	35.1 (30.3–36.7)	0.937
Emergent CS, *n* (%)	26 (47.3)	17 (48.6%)	9 (45.0%)	1.000
Operation time (minutes)	56.0 (30–95)	59.0 (30–95)	52.5 (35–95)	0.426
Estimated blood loss (mL)	1290 (400–3400)	1330 (400–3400)	1190 (560–3225)	0.618
Transfusion, *n* (%)	21 (38.2%)	11 (31.4%)	10 (50.0%)	0.282
Ward method, *n* (%)	17 (30.9%)	5 (14.3%)	12 (60.0%)	<0.001

CTG, cardiotocogram: GW, gestational week: CS, cesarean section. Data are given as median (range) or *n* (%).

**Table 4 diagnostics-11-01369-t004:** Neonatal outcomes.

	Total *n* = 58	Type 1 *n* = 38	Non-Type 1 *n* = 20	*p*-Value
Birth weight (gram)	2221 (1472–2940)	2173 (1472–2775)	2277 (1541–2940)	0.314
Standard deviations of birth weight	−0.02 (−1.30–1.34)	−0.05 (−1.30–1.34)	0.25 (−1.14–1.16)	0.239
Female/Male	27/31	20/18	7/13	0.316
Apgar score 1 min	8 (1–9)	8 (1–9)	8 (4–8)	0.178
Apgar score 5 min	9 (6–10)	9 (6–10)	9 (7–9)	0.850
pH of umbilical artery	7.299 (7.172–7.403)	7.308 (7.172–7.403)	7.279 (7.204–7.397)	0.086
Hemoglobin at birth (g/dL)	13.6 (10.9–18.4)	13.6 (10.9–17.4)	13.9 (11.6–18.4)	0.507
Respiratory support, *n* (%)	39 (70.9%)	26 (74.3%)	13 (65.0%)	0.674
Respiratory support (days)	3 (0–42)	3 (0–30)	3 (0–42)	0.913
Congenital disorders, *n* (%)	4 (6.9%)	2 (5.3%)	2 (10%)	0.428
Admission days	16 (6–73)	14 (6–59)	16 (6–73)	0.616

No neonate was necessitated with transfusion by exsanguination. Data are given as median (range) or *n* (%).

## Data Availability

The data presented in this study are available on request from the corresponding author.

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
