# Peer review of "Placental Types and Effective Perinatal Management of Vasa Previa: Lessons from 55 Cases in a Single Institution"

_diagnostics, 2021, doi:10.3390/diagnostics11081369_

Round 1

Reviewer 1 Report

I read with great interest the manuscript, which falls within the aim of this Journal. In my honest opinion, the topic is interesting enough to attract the readers’ attention. Nevertheless, authors should clarify some points and improve the discussion, as suggested below.

Authors should consider the following recommendations:

  • Manuscript should be further revised in order to correct some typos and improve style.
  • I would recommend to stress how placental diseases may arise, at leat in part, by epigenetic changes which may modify the placenta vascular framework development (authors may refer to: PMID: 28466013; PMID: 28282763).

Author Response

Dear Reviewer 1.

We appreciate for your thoughtful comments and advices to refine our manuscript.

Reviewer1

Following reviewer’s comments, we corrected some typos.

We also added two papers [38,39], which the reviewer suggested and we added the sentence in the line 313 to 312, as follows; ‘Recent advances of genome-wide studies of placental diseases may also add helpful information to elucidate the underlying mechanisms [38,39]. ‘

[Ref38]: Chiofalo B, Laganà AS, Vaiarelli A, La Rosa VL, Rossetti D, Palmara V, Valenti G, Rapisarda AMC, Granese R, Sapia F, Triolo O, Vitale SG. Do miRNAs Play a Role in Fetal Growth Restriction? A Fresh Look to a Busy Corner. Biomed Res Int. 2017; 2017: 6073167.

[Ref39]: Laganà AS, Vitale SG, Sapia F, Valenti G, Corrado F, Padula F, Rapisarda AMC, D'Anna R. miRNA expression for early diagnosis of preeclampsia onset: hope or hype? J Matern Fetal Neonatal Med. 2018; 31: 817-821.

We appreciate your kind consideration for our paper publication.

Reviewer 2 Report

I read with great interest the paper. I find it well wrote and with good idea research

Below my suggestions

  1. Introduction: Well add the placenta previa can be also risk factor for onset infection after Caesarean section expecially in low setting (see and cite Maternal caesarean section infection (MACSI) in Sierra Leone: a case-control study. Epidemiol Infect. 2020)
  2. Methods and results: are clear
  3. Discussion: add some proposal during screening and pre natal visit to reduce the burden and the consequence of this anatomical condition

Congratulations for your interesting paper

Author Response

Dear Reviewer 2.

We appreciate for your thoughtful comments and advices to refine our manuscript.

We added the sentence that we could not reduce the burden of prenatal visit in the limitation part. We also cited the suggested paper in the part of Discussion: line 319 to 321, as the reference number 40. We add the sentence in the limitation paragraph as follows; ‘In addition, we lack the proposal to reduce the burden of screening and prenatal visit, as well as the consequences of abnormal placentation leading to post-operative complications [40].’

[Ref40]: Di Gennaro F, Marotta C, Pisani L, Veronese N, Pisani V, Lippolis V, Pellizer G, Pizzol D, Tognon F, Bavaro DF, Oliva F, Ponte S, Nanka Bruce P, Monno L, Saracino A, Koroma MM, Putoto G. Maternal caesarean section infection (MACSI) in Sierra Leone: a case-control study. Epidemiol Infect. 2020; 27: 148e40. 

We appreciate your kind consideration for our paper publication.